# Implementation, Feasibility, and Acceptability of MATCH to Prevent Iatrogenic Disability in Hospitalized Older Adults: A Question of Geriatric Care Program?

**DOI:** 10.3390/healthcare11081186

**Published:** 2023-04-20

**Authors:** Eva Peyrusqué, Marie-Jeanne Kergoat, Marie-Josée Sirois, Nathalie Veillette, Raquel Fonseca, Mylène Aubertin-Leheudre

**Affiliations:** 1Centre de Recherche, Institut Universitaire de Gériatrie de Montréal, Montreal, QC H3W 1W4, Canada; eva.peyrusque@gmail.com (E.P.);; 2Département des Sciences de l’Activité Physique, Université du Québec à Montréal, Montreal, QC H3C 3P8, Canada; 3Faculty of Medicine, Université de Montréal, Montreal, QC H3T 1J4, Canada; 4Département de Réadaptation, Université de Laval, Quebec, QC G1V 0A6, Canada; 5Centre d’Excellence sur le Vieillissement de Québec, Quebec, QC G1S 4L8, Canada; 6Département de Sciences Économique, École des Sciences de la Gestion, Université du Québec à Montréal, Montreal, QC H2X 1L4, Canada

**Keywords:** frailty, geriatric unit, physical activity, hospital care, mobility

## Abstract

Senior adults (>age 65) represent almost 20% of the population but account for 48% of hospital bed occupancy. In older adults, hospitalization often results in functional decline (i.e., iatrogenic disability) and, consequently, the loss of autonomy. Physical activity (PA) has been shown to counteract these declines effectively. Nevertheless, PA is not implemented in standard clinical practice. We previously showed that MATCH, a pragmatic, specific, adapted, and unsupervised PA program, was feasible and acceptable in a geriatric assessment unit (GAU) and a COVID-19 geriatric unit. This feasibility study aims to confirm that this tool could be implemented in other geriatric care programs, notably a geriatric rehabilitation unit (GRU) and a post-acute care unit (PACU), in order to reach the maximum number of older patients. Eligibility and consent were assessed by the physician for all the patients admitted to the three units (GAU, GRU, and PACU). The rehabilitation therapist taught each participant one of the five PA programs based on their mobility score on the decisional tree. Implementation (eligibility (%): patients eligible/number admitted and delay of implementation: number of days until prescription); feasibility (adherence (%): number sessions completed/number sessions prescribed and walking time (%): total walking time/time prescribed time); and acceptability (healthcare team (%): tool adequacy (yes/no) and patient: System Usability Scale questionnaire (SUS: x/100)) were evaluated and analyzed using a Kruskal–Wallis ANOVA or Fisher’s exact test. Eligibility was different between the units (GRU = 32.5% vs. PACU = 26.6% vs. GAU = 56.0%; *p* < 0.001), but the time before implementation was similar (days: GRU = 5.91 vs. PACU = 5.88 vs. GAU = 4.78; *p* > 0.05). PA adherence (GRU = 83.5% vs. PACU = 71.9% vs. GAU = 74.3%) and walking time (100% in all units) were similar (*p* > 0.05). Patients (SUS: GRU = 74.6 vs. PACU = 77.2 vs. GAU = 77.2; *p* > 0.05) and clinicians (adequacy (yes; %): GRU = 78.3%; PACU = 76.0%; GAU = 72.2%; *p* > 0.05) found MATCH acceptable. Overall, MATCH was implementable, feasible, and acceptable in a GAU, GRU, and PACU. Randomized controlled trials are needed to confirm our results and evaluate the health benefits of MATCH compared with usual care.

## 1. Introduction

Hospitalization is associated with an increase in sedentary time (bed rest), which in older adults leads to a decline in muscle function (i.e., muscle strength: −8% [1]), physical performance (i.e., walking speed: −7.5% [1]), and activities of daily living (ADL: from −23% to −63% [2]). All these physical deteriorations related to hospitalization are collectively defined as iatrogenic disability [3]. At discharge, iatrogenic disability increases the risk of falls (+14% to +34% [4]), re-admission (+33%), home-based services, and institutionalization [5]. Finding solutions to counteract this vicious cycle is important because it could potentially improve the quality of life and reduce the risk of mortality and the burden of healthcare costs [6,7,8,9].

Fortunately, it has been reported that physical activity (PA) can counteract the vicious iatrogenic disability cycle, especially when prescribed within the first days of hospitalization [10]. A recent meta-analysis confirmed that older patients who received an exercise intervention (group and/or supervised sessions) improved their walking speed and shortened their length of stay (LOS), compared with the control group [11]. A recent randomized controlled trial (RCT) performed in an acute care unit suggests that an individual supervised exercise intervention (daily moderate-intensity session including resistance (weight training with equipment), balance, and walking exercises), improved physical performance (SPPB score: +2.2 pts [12]), and muscle function (power and strength [13]) compared with the usual care group. Another study highlighted that patients admitted into an acute care unit and performing a supervised group exercise intervention (weekdays; two exercises/session: walking and sit-to-stand exercises) improved their ADL levels compared with the usual care group (OR: 0.32 [14,15]).

Despite the well-established health benefits of PA, it still needs to be fully integrated into usual care (real-life settings), including various geriatric care units. Barriers to promoting PA as routine care for hospitalized older adults include (1) clinicians’ attitude and awareness (ageism, behaviour) and lack of knowledge; (2) patients’ level of fitness (intrapersonal level); (3) safety concerns; (4) lack of space and resources (including time or equipment; institutional level); and (5) the need for well-defined PA protocols [16]. Thus, to counteract these barriers, geriatric clinicians (MDs, nurses, kinesiologists, or physiotherapists) and researchers in gerontology collaborated to establish a pragmatic exercise intervention known as MATCH (Maintenance of AuTonomy through exerCise during Hospitalization) using a co-creation design to implement PA as usual care during hospitalization. MATCH included five simple, individual, unsupervised, and adapted PA programs. Their prescription was based on the patient mobility profile score obtained using a decisional tree [17]. We previously observed that MATCH was safe (no falls), feasible, and acceptable (i.e., adherence: 66 and 53% of the prescription performed, respectively; patient satisfaction: 91% and 82%, respectively) in a geriatric assessment unit (GAU [18]) and a COVID-19 geriatric unit [19]. The GAU did not include rehabilitation care, and the COVID-19 geriatric unit did not allow group or supervised interventions due to public health restrictions. However, before generalizing this intervention (MATCH) and implementing exercise as usual care, evaluating it in more common geriatric care programs is essential. Thus, this single-arm feasibility study aimed to assess MATCH’s implementation, feasibility, and acceptability in a geriatric rehabilitation unit (GRU) and a post-acute care unit (PACU). These two units were selected because they included rehabilitation time in usual care (evaluating the burden of adding exercise as part of care) and a geriatric population outside of geriatric care programs from various healthcare systems. Overall, this study is important since 20% of the population is over age 65 [20] but account for 48% of hospital bed occupancy [21]. In addition, hospitalization due to iatrogenic disability further increases the healthcare burden of this population.

## 2. Materials and Methods

### 2.1. Design

This feasibility study was conducted as a single-arm trial at a single geriatric hospital (the Institut Universitaire de Gériatrie de Montréal (IUGM)) and included patients admitted to three different geriatric programs (three GPs: GAU, GRU, and PACU).

### 2.2. Participants

#### 2.2.1. Changes due to the COVID-19 Pandemic

The study was conducted during the COVID-19 pandemic. Due to the unexpected and uncontrolled COVID-19 restrictions, the recruitment flow was intermittently interrupted. Moreover, during the pandemic, the 3 GPs were merged and not separated in a dedicated space (i.e., hospital floor).

#### 2.2.2. Participants’ Consent

The physician screened all the patients admitted to the 3 GPs within 24–48 h (or post-delirium) to determine their eligibility (based on the inclusion and exclusion criteria). Eligible patients provided verbal informed consent to accept or decline care (MATCH).

#### 2.2.3. Inclusion and Exclusion Criteria

The selection criteria were patients (1) aged >65 years old; (2) able to understand, speak, and read French or English; (3) not living in a nursing home; (4) physically able to take part in a PA program; and (5) with a Mini-Mental State Examination score ≥18/30 and/or presence of self-criticism. The exclusion criteria were (1) being in the terminal phase of life, (2) unable to collaborate, (3) having delirium, (4) having suffered a recent fracture that does allow them to exercise, (5) having chronic pain, (6) having a short hospital stay (<5 days), and (7) having visual and/or hearing impairments.

### 2.3. MATCH Tool and Intervention

The MATCH tool included a decisional tree and five related coloured exercise programs (from red for very frail patients with low mobility to blue for more physically independent ones [17,18]). The decisional tree incorporated three standard validated geriatric tests: (1) a 30 s chair test [22], (2) a side-by-side and semi-tandem balance test [23], and (3) a 4 m walking speed test [24]. The score from the first two tests determined the prescribed program, which included two specific and adapted exercises. The third test determined the prescribed walking time for all levels, except the red program.

All MATCH programs were created to improve balance, mobility, and muscular function through functional exercises (seated knee extension, the squeezing hand exercise, sit-to-stand exercises with or without support, side walking in the upright position, knee raise in sitting and standing positions with support, and wall half-squat (video and tool pamphlet are available in the following website: https://www.trainingrecommend.com/ (accessed on 5 January 2023), and walking time (from 10 to 30 min/day).

All the participants were asked to perform each program thrice daily, even if the target was set to at least two sessions per day (based on the 150 min/week of PA to be considered active). The prescription was similar between the participants even if the rehabilitation care was different in the 3 GPs (GRU: ~5 times/week vs. PACU: from 3 to 4 times/week vs. GAU: from 0 to 4 times/week) since MATCH was implemented to limit sedentary time outside rehabilitation care.

The functional exercises were performed unsupervised, without specific material, in the patient’s room, and at the time of day of their choice. The walk was performed unsupervised in the hallway or in the patient’s room in case of a COVID-19 outbreak. Participants could complete their exercises or walking prescriptions using mobility aids, such as a walker or cane, to ensure safety.

### 2.4. MATCH Implementation Procedures (Figure 1)

Patients who were eligible and consented to participate were evaluated by a rehabilitation therapist using the decisional tree (48–72 h post-admission). Thereafter, the physician prescribed the MATCH program related to the decisional tree score, and the rehabilitation therapist taught the patient the prescribed exercises during one session. Following the training and until discharge, the patient was expected to perform the MATCH prescription. Finally, the physician followed adherence during their daily visit and provided generated feedback according to their answer.

**Figure 1 healthcare-11-01186-f001:**
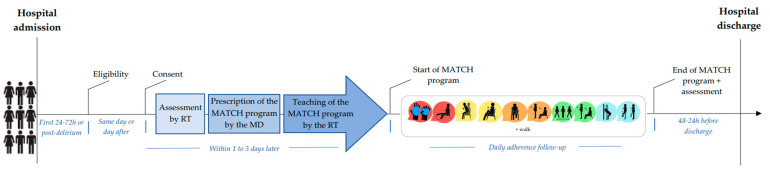
Timeline of the study protocol. Legend: MD = physician; RT = rehabilitation therapist.

### 2.5. Measures

General characteristics: Medical records were used to characterize the population (age, gender, and Mini-Mental State Examination (MMSE; total score from 0 to 30)). The cut-off was chosen via a medical consensus (pragmatic design) based on the Folstein Scoring Scale [25], the Geriatric Depression Scale-4 (GDS-4; total score from 0 to 4 where 0 indicates no depression, and ≥1 indicates extreme suspicion of depression [26]), LOS, walking speed, body mass index (BMI), and rehabilitation care time.

MATCH tool implementation: Hospital implementation was estimated using the ratio between the number of patients evaluated by the physician and the number of patients eligible to participate in the project (eligibility rate; %), and the ratio between the number of patients included in the project and the number of eligible patients (inclusion rate; %). Data on the level of difficulty (%) of each PA program prescribed and delay in implementation (days between admission and teaching session) were also collected.

MATCH tool feasibility: Participants were asked to record the number of sessions completed each day in a logbook. Feasibility was calculated using adherence throughout the intervention (ratio between the number of completed exercise sessions and the number of prescribed exercise sessions (%) + ratio between walking time performed/walking time prescribed (%)) among participants who completed the evaluation (per-protocol analysis).

MATCH tool acceptability: -Healthcare team acceptability was assessed by asking them if, according to their clinical judgement, they felt that the program prescribed was appropriate for the patient (“yes” or “no”);-Patient acceptability was determined before discharge using the System Usability Scale (SUS) questionnaire and the enjoyment Likert scales. The SUS questionnaire included 10 questions with scores between 0 (not satisfied) and 100 (very satisfied) [27]. The score must be >71.4/100 to be considered acceptable [28]. The enjoyment Likert scale included four answers from “strongly disagree” to “strongly agree”.

### 2.6. Sample Size

#### 2.6.1. Recruitment Timeline

Recruitment was carried out from October 2019 to August 2022. However, due to the COVID-19 pandemic restrictions, our recruitment was interrupted as follows: 1st wave = March to August 2020; 2nd wave = December 2020 to April 2021. Given the design of this single-arm feasibility study, all the patients admitted to the 3 GPs during these different periods, were assessed for eligibility. There is no consensus on the required sample size for feasibility studies, and recommendations vary from 10–12 to 60–75 per group, depending on the study objectives. Based on a previous study by Lewis et al. [29], the sample size (two-arm parallel design) including hypothesis testing (*α* = 0.05, 1 − *β* = 0.90) for 3 criteria is defined as follows: a total of 78 screen patients are required for recruitment uptake ≥35%, 34 patients/group are required for adherence to intervention ≥75%, and 22 patients/group are required for follow-up ≥85%.

#### 2.6.2. End of the Study

The recruitment phase was concluded at the end of the planned 12-month period, which was necessary to achieve the primary objective of the study: evaluating the feasibility and acceptability of tool implementation (Phase II).

### 2.7. Statistical Analysis

Quantitative data were expressed as means ± SD, whereas qualitative data were expressed as percentages. The normality of data was verified graphically and through the Shapiro–Wilk test and homoscedasticity using Levene’s test. Given the non-normal distribution of the data and the small number of participants per group, we performed nonparametric statistical tests. General patient characteristics and implementation, feasibility, and acceptability variables of the three groups were compared using a Kruskal–Wallis (nonparametric) test for quantitative data or Fisher’s exact test for qualitative/categorical data. We first tested the null hypothesis of equality between groups for all comparisons. When such tests were significant, post hoc Mann–Whitney tests were used to identify the differences between groups (2 by 2).

The effect size for quantitative data was calculated using Eta squared (η^2^), which was obtained using the following formula: η^2^(H) = H − (k − 1)/(*n* − k), where H is the Kruskal–Wallis statistic, k is the number of groups, and *n* is the number of participants. The resulting value of η^2^(H) can be interpreted as a small (0.01), medium (0.06), or large (0.14) effect size. Concerning the effect size for categorical data, Phi coefficients (φ) were computed. The resulting value of φ can be interpreted as a small (0.1), medium (0.3), or large (0.5) effect size [30]. As we used a per-protocol analysis (feasibility to implement MATCH), only participants who completed the pre- and post-intervention evaluations were included in the analyses. IBM SPSS Statistics for Windows, version 28.0 (SPSS Inc., Chicago, IL, USA) was used, and *p* < 0.05 was considered significant.

## 3. Results

### 3.1. Participants

A total of 519 patients were admitted to the 3 GPs, and 181 were eligible (34.8%), as shown in Figure 2. Among the eligible participants, 147 consented to participate and underwent a baseline evaluation (81.2%). Finally, 104/107 patients completed the post-intervention evaluation (70.7%; see flowchart in Figure 2 for more details).

### 3.2. Participant Characteristics

Baseline characteristics such as age, sex, cognitive status, and BMI were similar for the three groups (*p* ≥ 0.05), as shown in Table 1, except for depression status (GRU group: 4.4% vs. PACU group: 20% vs. GAU group: 50%; *p* < 0.001), walking speed (GRU group: 0.44 ± 0.16 m/s vs. PACU group: 0.45 ± 0.18 m/s vs. GAU group: 0.64 ± 0.29 m/s; *p* < 0.05), diagnosis (*p* < 0.001), rehabilitation care time (*p* < 0.001), and LOS (*p* < 0.05). These differences are due to the different assignments in each GP.

### 3.3. MATCH Tool Implementation

The implementation time (delay), LOS, and percentage of LOS with MATCH are described in Table 2. Overall, no differences were observed between the three groups (*p* ≥ 0.05).

Figure 3 details the MATCH PA program prescribed to patients in each unit (GRU, PACU, and GAU). Briefly, MATCH PA programs were distributed as follows: red (7%, 0%, and 6%, respectively); yellow (15%, 24%, and 22%, respectively); orange (61%, 44%, and 17%, respectively); green (13%, 24%, and 39%, respectively); and blue (4%, 8%, and 17%, respectively). A difference was observed between the GRU and GAU groups regarding the distribution of the prescribed MATCH PA programs (*p* = 0.007; φ = 0.44).

The prescription for the walking program varied from 0 to 30 min. Differences were observed between GRU and GAU groups (*p* < 0.001) as well as between PACU and GAU groups (*p* = 0.003; φ = 0.55) (see Table 3).

### 3.4. Feasibility of MATCH Implementation

Overall, the prescribed exercises were performed twice per day during at least 71.9% of the LOS (GRU = 83.5%; PACU = 71.9%; GAU = 74.3%) and three times per day for at least 59.6% of the LOS (GRU = 71.9%; PACU = 59.6%; GAU = 67.5%). No difference was observed between the groups (*p* > 0.05; η^2^(H) = 0.025). The walking goal was fully completed for the three groups (100% of the prescription; *p* = 1.00; η^2^(H) = −0.021).

### 3.5. Acceptability of MATCH Implementation

The rehabilitation therapists considered the prescribed program adequate for 78.3% of the patients in the GRU group, 76% in the PACU group, and 72.2% in the GAU group (*p* = 0.94; φ = 0.46). In addition, 91.2% of the patients in the GRU group enjoyed or enjoyed the program a lot, as well as 88%of patients in the PACU group and 94.4% in the GAU group (*p* = 0.46; φ = 0.26). Finally, the mean score on the SUS questionnaire was similar (GRU: 74.6%, PACU: 77.2%, and GAU: 77.2%; *p* = 0.76; η^2^(H) = −0.02)) and reached the acceptability threshold (>71.4%) for the three groups.

## 4. Discussion

The main purpose of this single-arm feasibility study was to assess the implementation, feasibility, and acceptability of MATCH, an unsupervised PA tool, in three different GPs.

First, the eligibility rate was significantly higher (*p* < 0.001; φ = 0.21) in the GAU group (56%) than in the GRU (32.5%) and PACU (26.6%) groups. These differences could be explained by the unit assignment and patient profiles in the GAU group [31]. Indeed, most of the time, the patients admitted to the GAU came from home, whereas those admitted to the PACU or GRU generally came from another hospital. Thus, those in the latter group were generally more disabled, with a slower walking speed (*p*-value > 0.05). Even if it was not statistically significant, the inclusion rate was higher for the GAU group (GAU: 86.3% vs. GRU: 80.9% vs. PACU: 75.6%; *p* = 0.22), as well as the difficulty level of the PA program prescribed (GAU: blue and green programs = 56%; *p* > 0.05; see Figure 3). However, the inclusion rate observed in this study is comparable to other studies that also added simple exercises to usual care (from 13.7% to 40.8% [32,33]). Thus, our results could be replicated in other hospital settings.

Second, it is recognized that mobilizing older adults as early as possible helps counteract or reduce iatrogenic decline [10]. Indeed, Hauer et al. reported that implementing a PA program immediately after prescription leads to a one-point improvement in the SPPB, which is considered clinically significant. In contrast, those who started the same PA program later gained less [10]. The relevance of PA programs is greatly influenced by their ability to be implemented in a timely manner in real-world settings. In this study, the time needed from admission to MATCH implementation varied from 4.8 (GAU) to 5.9 days (GRU). However, about LOS, the patients in all care units spent the same amount of time performing the PA program (GRU = 79.4% vs. PACU = 83.1% vs. GAU = 82.8% of hospitalization). This shorter implementation time for the GAU group could be explained by the unit assignments and patient profiles in these units (home vs. hospital transfer). The delay was longer than previously observed (~5.5 vs. 3 days), which could be due to the lack of human resources/shortage of staff or public health restrictions during the COVID-19 pandemic [18] but could be considered efficient to reduce iatrogenic disability.

Moreover, all levels of MATCH exercises were prescribed in all GPs (except the red program for the PACU group; see Figure 3). This means that almost all patients could benefit from an adapted PA program regardless of their physical condition (high mobility level (blue) to low mobility level (red)). This result is important, as independent or very frail patients are rarely included in the priority of healthcare teams [34,35]. In addition, it has been shown that healthcare teams are less likely to encourage this type of care in physically independent patients [35]. However, very frail patients are highly represented in hospitalized older adults. Thus, having a pragmatic exercise tool that is more inclusive should be considered relevant, as it helps healthcare teams to prevent iatrogenic disability in patients, as well as its consequences and burden.

Furthermore, adherence to the MATCH tool was considered good to very good even if unsupervised. Indeed, the walking prescription was performed 100% of the time, and the prescribed exercises were performed twice daily during at least 71.9% of the LOS. The literature suggests that lack of personnel, time, and equipment are among the main barriers preventing the practice of PA in hospitalized patients [36,37,38]. Therefore, the MATCH tool can overcome these barriers, as patients can perform the exercises whenever they want, without specific gym equipment or space.

Interestingly, even if the participants from the GRU had a rehabilitation session every weekday, this group had greater but nonsignificant adherence compared with the other groups (GAU and PACU). The participants in the three groups exercised 30 min per day on average and reached the PA recommendation of the American College of Sports Medicine (>150 min/week [39]), even if each MATCH program does not require professional supervision. A systematic review noted that hospitalized patients spent 93% to 98.8% of the time (i.e., approximately 23 h per day) in a sedentary position and walked less than 1000 steps per day [40]. Another study, which evaluated 24 h of mobility during acute hospitalization, confirmed that older patients spent 17 h per day in bed [41]. Therefore, our results show that the MATCH tool can help older adults stay physically active during hospitalization, and PA can be practiced independently from the care programs.

In addition, the MATCH tool was also deemed acceptable by patients and healthcare professionals in the three geriatric units. Indeed, almost all patients (from 88% to 94.4%) enjoyed or very much enjoyed the program. The PACU group had the lowest score even if it was considered very good (88%). This finding is consistent with the fact that this group (PACU) had the lowest completion of the exercises. A similar study showed that only 70% of patients enjoyed performing their PA intervention [42]. In addition, another study reported that enjoyment and motivation were higher for people who completed their exercises using instruction leaflets (internal) than those using exergames (external) during 10 days of hospitalization [43]. This difference can be explained by the type of motivation (external vs. internal), which is an important element in acceptability and adherence to a PA practice [44]. Thus, interventions using intrinsic motivation (self-determination as with MATCH) should be considered a key element for exercise tool implementation.

Finally, patients from the three units reported being very satisfied with MATCH as a care tool (SUS score > 71.4% [28]). Rehabilitation therapists also deemed the program adequate for their patients (72.2% to 78.3%). The main qualitative reasons reported for inadequate program implementation were in cases of chronic or acute pain (*n* = 6) or a specific disease (stroke or Parkinson’s disease; *n* = 4). These results are similar to those found in our previous pilot study in a GAU, carried out before the COVID-19 pandemic [18]. These results show that the MATCH tool is acceptable to patients and rehabilitation therapists, independently of the care programs, which is an essential factor for the sustainability of a new care.

This study has some limitations. The number of participants, which was relatively small within each group due to COVID-19 protocol changes and the absence of a control group due to the study design (single-arm feasibility study; Phase II), led to the lack of significance (unpowered study). However, this is not uncommon in feasibility studies. In addition, the recruitment period was performed during the height of the COVID-19 pandemic. Thus, the care conditions during these periods were not usual (public health restrictions, etc.). Finally, the Canadian healthcare system is different than in many other countries. Therefore, it is not possible to generalize our findings. Thus, powered RCTs (Phase III) are needed in other healthcare systems to confirm our feasibility conclusion and examine the efficacy of MATCH outside the pandemic context.

## 5. Conclusions

Implementing MATCH, an unsupervised and simple PA program, seemed feasible and acceptable not only for older patients hospitalized in a GRU or PACU but also for those in a GAU during COVID-19 public health restrictions. In addition, implementing MATCH appears to allow the healthcare team to counteract bed rest since the participants reached the weekly PA recommendation. Having a tool that can be implemented in all geriatric care programs and overcomes barriers will help reach as many patients as possible to improve healthcare outcomes in older adults. However, RCTs (Phase III) are needed to confirm our promising results and explore the efficacy of implementing a PA program on physical health compared with usual care.

## Figures and Tables

**Figure 2 healthcare-11-01186-f002:**
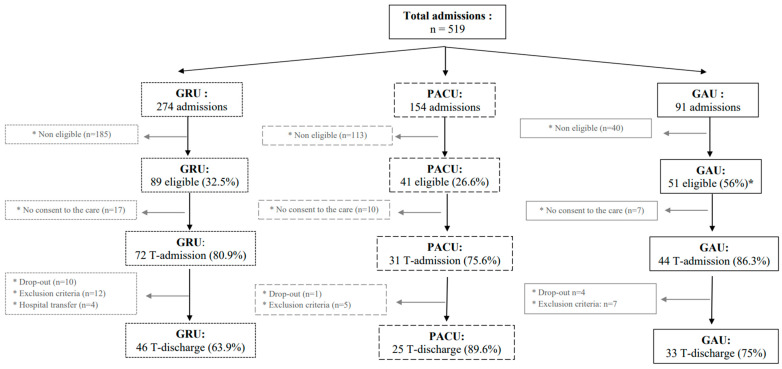
Flowchart. Legend: GRU = geriatric rehabilitation unit; PACU = post-acute care unit; GAU = geriatric assessment unit; T = measuring time. A significant between-group difference was found for the percent of eligible participants (* *p* < 0.001; φ = 0.21); no significant difference was found between groups for the inclusion rate (*p* = 0.42; φ = 0.10) and participants involved between T-admission and T-discharge (*p* = 0.19; φ = 0.15).

**Figure 3 healthcare-11-01186-f003:**
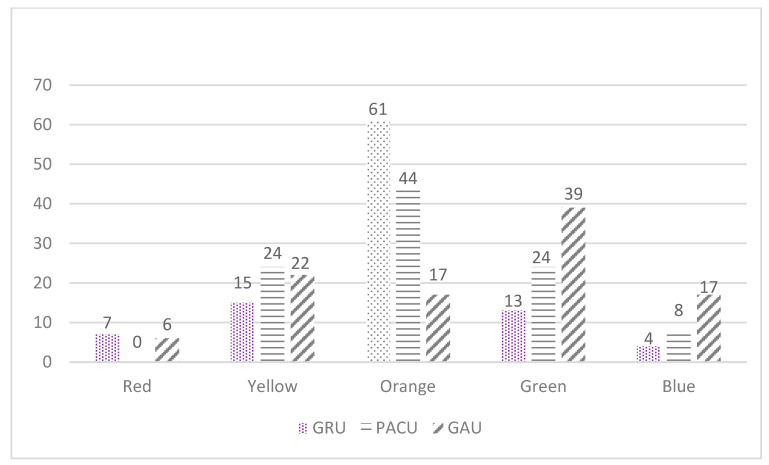
Distribution of MATCH PA program prescription (%). Legend: GRU = geriatric rehabilitation unit; PACU = post-acute care unit; GAU = geriatric assessment unit.

**Table 1 healthcare-11-01186-t001:** Baseline sociodemographic characteristics of participants.

Variables	GRU	PACU	GAU	*p*-Value (ES)
Age (years) [95% CI]	82.4 ± 7.9 [80.1–84.8]	83.8 ± 7.2 [80.8–86.8]	79.5 ± 7.6 [75.8–83.3]	0.24 ^†^ (0.03 ^a^)
Women (*n*; (%))	23 (50%)	15 (60%)	10 (55.5%)	0.75 ^§^ (0.09 ^b^)
Cognitive Status (*n*; (%))				0.34 ^§^ (0.2 ^b^)
Good cognition (Good self-criticism or MMSE ≥ 22)	38 (82.6%)	22 (88.0%)	18 (100%)	
Slightly impaired cognition (Slightly impaired self-criticism or MMSE = 18–21)	7 (15.2%)	3 (12.0%)	0 (0%)	
Impaired cognition (Impaired self-criticism or MMSE < 18)	1 (2.2%)	0 (0%)	0 (0%)	
Geriatric Depression Scale-4 (*n*; % of depression)	2 (4.4%) ^$^	5 (20.0%) ^&^	9 (50.0%) ^$&^	<0.001 ^§^ (0.45 ^b^)
Body mass index (kg/m^2^) [95% CI]	25.9 ± 4.6 [24.6–27.3]	23.5 ± 4.3 [21.8–25.3]	24.6 ± 5.3 [21.9–27.2]	0.054 ^†^ (0.08 ^a^)
Walking speed (m/s) [95% CI]	0.44 ± 0.16 ^$^ [0.39–0.49]	0.45 ± 0.18 ^&^ [0.37–0.53]	0.64 ± 0.29 ^$&^ [0.50–0.78]	<0.05 ^†^ (0.12 ^a^)
Diagnosis (*n*; (%))				<0.001 ^§^ (0.54 ^b^)
Neurological	18 (39%) ^$#^	1 (4%) ^&#^	1 (5.5%) ^$&^	
Traumatology	4 (9.0%)	10 (40.0%)	1 (5.5%)	
Deconditioning	24 (52.0%)	14 (56.0%)	16 (89%)	
Rehabilitation care time (min/working day) [95% CI]	33.0 ± 9.0 ^$#^ [30.3–35.8]	26.3 ± 9.7 ^&#^ [22.3–30.3]	12.6 ± 10.1 ^$&^ [6.4–18.7]	<0.001 ^†^ (0.27 ^a^)
Rehabilitation care time (min/working and weekend days) [95% CI]	24.6 ± 6.8 ^$#^ [22.6–26.7]	19.8 ± 7.4 ^&#^ [16.7–22.8]	9.6 ± 7.6 ^$&^ [4.9–14.2]	<0.001 ^†^ (0.28 ^a^)
Length of stay (working days) [95% CI]	29.2 ± 12.8 [25.4–33.0]	28.1 ± 11.1 [23.5–32.7]	22.9 ± 9.3 [18.3–27.6]	0.05 ^†^ (0.07 ^a^)
Length of stay (working and weekend days) [95% CI]	39.4 ± 17.9 ^$^ [34.1–44.7]	37.6 ± 15.4 [31.2–43.9]	30.3 ± 13.3 ^$^ [23.7–36.9]	0.043 ^†^ (0.06 ^a^)

Legend: *p*-value obtained using nonparametric ANOVA ^†^ (Kruskal–Wallis) and Fisher Test ^§^ (dichotomic variable); data are presented as % or mean ± SD [95% CI]; *p* < 0.05 significant. ^&^ = significantly different between PACU and GAU; ^$^ = significantly different between GRU and GAU; ^#^ = significantly different between GRU and PACU; effect size for Kruskal–Wallis test obtained using η^2^(H) ^a^ and for Fisher test obtained using φ ^b^. ES = effect size; GRU = geriatric rehabilitation unit; PACU = post-acute care unit; GAU = geriatric assessment unit. Working day = weekdays excluding weekends and public holidays.

**Table 2 healthcare-11-01186-t002:** MATCH tool implementation.

Variables	GRU	PACU	GAU	*p*-Value (ES)
Implementation time (weekdays) [95% CI]	5.9 ± 2.3 [5.78–7.74]	5.9 ± 3.3 [4.52–8.36]	4.8 ± 1.2 [3.76–6.02]	0.22 (0.024)
Implementation time (weekdays and weekends) [95% CI]	6.8 ± 3.0 [5.22–6.60]	6.4 ± 4.6 [4.5–7.26]	4.9 ± 2.3 [4.17–5.38]	0.14 (0.018)
% of LOS with MATCH (weekdays) [95% CI]	76.2 [72.3–80.0]	78.8 [74.9–82.6]	77.3 [73.6–81.0]	0.87 (0.017)
% of LOS with MATCH (weekdays and weekends) [95% CI]	79.4 [75.6–83.1]	83.1 [79.0–87.2]	82.8 [78.8–86.9]	0.55 (−0.020)

Legend: *p*-value obtained using a nonparametric ANOVA (Kruskal–Wallis); data are presented as follows: % or mean ± SD [95% CI]; *p* < 0.05 significant. Effect size for Kruskal–Wallis test obtained using η^2^(H). ES = effect size; GRU = geriatric rehabilitation unit; PACU = post-acute care unit; GAU = geriatric assessment unit; LOS = length of stay; % of LOS with MATCH was calculated as follows: 100 − ((delay implantation × 100)/LOS); weekdays = weekdays excluding weekends and statutory holidays for professionals.

**Table 3 healthcare-11-01186-t003:** Distribution of the MATCH walking time prescription (%).

	0-min	10 min	15 min	20 min	30 min
GRU ^$^	4	30	46	9	11
PACU ^&^	0	20	48	16	16
GAU ^&$^	0	17	17	0	66

Legend: GRU = geriatric rehabilitation unit; PACU = post-acute care unit; GAU = geriatric assessment unit; *p* < 0.05 significant. ^&^ = significantly different between PACU and GAU; ^$^ = significantly different between GRU and GAU.

## Data Availability

The datasets used in the current study are available from the corresponding author upon reasonable request.

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
