# Peer review of "Implementation, Feasibility, and Acceptability of MATCH to Prevent Iatrogenic Disability in Hospitalized Older Adults: A Question of Geriatric Care Program?"

_healthcare, 2023, doi:10.3390/healthcare11081186_

Round 1

Reviewer 1 Report (Previous Reviewer 1)

The autors have improved the ms

Author Response

Thank you for your time and your feedback.

Reviewer 2 Report (New Reviewer)

1. What training techniques are applied in this MATCH program? Can you describe one by one?

2. What do you mean by physically able to take part in a PA program, is it possible for patients with the use of walking aids such as walkers?

Author Response

Thank you for your time and review. Attached are the answers to your 2 points. 

Kind regards.

Reviewer 3 Report (New Reviewer)

I strongly recommend that authors completely rewrite the abstract, as it is presented right now it does not clearly provide the motivation and overall results of the work. Further, the introduction begins with several numerical values that are not previously introduced, this makes it difficult to understand.

The introduction section does not provide the sufficient background to fully understand the motivation of this work. Previous results should be clearly discussed in this section.

Section 2.1 should be expanded as it does not provide sufficient information to stand as its own within the Materials and methods as it is presented right now.

Enhanced the quality of Figure 1, and accurately place it after it is mentioned.

Section 2.6 is regarding the sample size which is not given n = ?

Table 1 should include the corresponding confidence intervals, and null hypothesis is not discussed.

Overall, this manuscript seems very rushed, each section has several flaws regarding the information being presented.

Authors should improve and rewrite the way they are presenting they results, concerning the statistics values presented, authors need to further explain what does imply for  a result to be ‘statistically significant’ in the scope of their research.

Author Response

Attached our revisions to your comments.

Kind regards.

Round 2

Reviewer 3 Report (New Reviewer)

One detail, it should be 95% CI, not 95% IC.

I strongly recommend to authors to make efforts in providing a video abstract explaining the MATCH tool. I really believe this could make your research more approachable not only to physicians but to the people in general, lower income countries could really benefit with this kind of research and information for older adults.

Author Response

Thank you for your eye, I modified "IC" for "CI".

We have already created a website that included all the videos for the exercises and the decisional tree. all this info are avialable here: https://www.trainingrecommend.com/.

As requested, we added this information in the section #2.3

We hope that this addition fully answers to your request

Thank you for your time.

This manuscript is a resubmission of an earlier submission. The following is a list of the peer review reports and author responses from that submission.

Round 1

Reviewer 1 Report

Implementation, feasibility, and acceptability of MATCH to prevent iatrogenic decline during hospitalization of the older adults: A question of geriatric care program?

Title:

-       No information about the study location and study design.

Abstract

-       Please mention the novelty of the study.

-       No information about the study method, including study design & analysis.

-       No feasibility results nor the outcome did not clearly state.  

Introduction

-       Need information about the definition of “iatrogenic function” and the description of the connection between iatrogenic function and older adult issues.

-       What is MATCH? It is important to describe the MATCH in the deep and detail in the introduction to show the novelty and its scientific reason. The author only emphasized the PA. It also did not clearly state the connection with the iatrogenic function.

-       The introduction section must be reorganized significantly. I suggest starting with the older adult -> connection with the iatrogenic function (emphasized the main variable or outcome and iatrogenic function)-> what the researcher did in the previous studies with this issue (emphasized the main variable or outcome and iatrogenic function) -> the author promotes a novel problem solving (MATCH) (emphasized the main variable or outcome and iatrogenic function). Mention its novelty- > hypothesis or aim.

Method

-       Please clearly state the intervention period, data collection, or follow-up time. It is better if the author provides the study design graphic.

-       No information on the study location.

-       Since the setting was in three different locations, how does the author make a standard in the intervention implementation?

Discussion

-       Since this study aims to evaluate the feasibility that utilizing the exploratory single-arm study, thus the author needs to provide the limitation of the study. It needs to present the potential idea in this issue for future study.

Author Response

A linguistic revision was made by a native translator.

Kind regards.

Reviewer 2 Report

Implementation, feasibility and acceptability of MATCH to prevent iatrogenic decline during hospitalization of the older adults: A question of geriatric care program?

The manuscript aimed to evaluate the feasibility and acceptability of implementing MATCH in different Geriatric Programs (GP).

The theme is relevant and I appreciate the efforts of the authors to conduct the study and write the text. Minor corrections are needed to make the text ready for publication.

Introduction

It is well written and addresses the importance of the research and the gaps in the literature.

Materials and methods

-     -  Lines 125-127: I would like to see more information about the cutoff scores of MMSE and GDS that you use to classify the participants into categories.

Results

-        -  I think the first paragraph (lines 154-56) is not part of the results of your manuscript.

-         - The variable “body mass index” was not mentioned in the methods section. Please, insert it in the methods section.

-       -   Table 1: the % of the cognitive status are not properly arranged. The last category (Impaired cognition (Impaired self-criticism or MMSE<18) is without frequencies.

-          Line 171: Please, correct “Baseline The implementation”.

Discussion

-          Lines 245 and 249: what does IFRU means?

-          Lines 261-262: you say that “Concerning the prescription of the MATCH exercises, all levels were prescribed in all GP (see Figure 2)”. However, the level red was not prescribed in the PACU GP.

-          Line 290: 94,4% (please, fix it – 94.4%)

References

-          The references are recent and relevant

Author Response

Kind regards.

Reviewer 3 Report

Thank you for the opportunity to get acquainted with your manuscript. I agree that the health problem of people of retirement age is a serious problem of modern medicine, which has a significant impact on the health of younger citizens. However, some points are not entirely clear in the article: how was the selection of subjects carried out? have any selection criteria been applied other than age and consent to participate? The fact is that low mobility can be due to various reasons: if the reason is general weakness, then walking can have a beneficial effect. But if a geriatric patient is sedentary due to acute pain, for example, arthrosis, then the solution to the problem is not so clear. I would also like to clarify the nature of walking as a method of therapy: whether it was walking in a hospital room or in a park. Were the classes group or individual? The exercise program mentioned in the text of the article also leaves many questions. I couldn't find any information about it. I think the article would look more convincing if a list of exercises were given. How innovative is this program? Have the author or other researchers carried out measures to test it earlier? It is well known that physical therapy, terrincourt and just walking are an integral part of the therapy of many geriatric patients. The article can be published after making changes

Author Response

Kind regards.

Round 2

Reviewer 1 Report

Dear Editor, thank you for the invitation. My decision is still the same. The author failed to provide a high rating for the study, particularly the statistical analysis. 

Thank you.